# Antibodies from the Sera of Multiple Sclerosis Patients Efficiently Hydrolyze Five Histones

**DOI:** 10.3390/biom9110741

**Published:** 2019-11-15

**Authors:** Svetlana V. Baranova, Elena V. Mikheeva, Valentina N. Buneva, Georgy A. Nevinsky

**Affiliations:** Institute of Chemical Biology and Fundamental Medicine, SD of Russian Academy of Sciences, 630132 Novosibirsk, Russia; swb@ngs.ru (S.V.B.); korobova_lena@mail.ru (E.V.M.); buneva@niboch.nsc.ru (V.N.B.)

**Keywords:** human blood antibodies, multiple sclerosis patients, catalytic IgGs, hydrolysis of human histones

## Abstract

It is known that intranuclear histones can be pernicious after entering to the extracellular space. In addition, the immunization of animals with exogenous histones leads to systemic inflammatory and toxic reactions. Abzymes—autoantibodies with enzymatic activities—are the distinctive feature of autoimmune diseases and they can be especially dangerous to humans. Here, electrophoretically homogeneous IgGs were isolated from sera of patients with multiple sclerosis (MS) by chromatography on several affinity sorbents. We present evidence that sera of all MS patients contain autoantibodies against histones and 73% of IgGs purified from the sera of 59 MS patients efficiently hydrolyze from one to five histones: H1, H2a, H2b, H3, and H4. The relative average efficiency of the histones hydrolysis was ~3.9-fold higher than that for healthy donors. The relative average activity of IgGs depends on the type of MS and decreased approximately in the following order: debut of MS, secondary progressive multiple sclerosis, remitting multiple sclerosis, remittent progressive multiple sclerosis. Similar to proteolytic abzymes of patients with several autoimmune diseases, histone-hydrolyzing IgGs from MS patients were inhibited in the presence of specific inhibitors of serine and of metal-dependent proteases, but an unexpected significant inhibition of the activity by inhibitors of thiol-like and especially acidic proteases was observed. Since IgGs can efficiently hydrolyze histones, a negative role of abzymes in the development of MS cannot be excluded.

## 1. Introduction

Abzymes (Abzs)—antibodies to chemically stable analogs to transition chemical states of different reactions—were well described (reviewed in [1,2,3]). During three the last decades, it has become clear that autoantibodies (auto-Abs) from sera of patients with different autoimmune pathologies can possess enzymatic activities [3,4,5,6,7,8]. Similarly to artificial abzymes against analogs of transition states of chemical reactions [1,2,3], naturally abzymes of autoimmune (AI) patients may be Abs synthesized by lymphocytes directly against different enzyme substrates acting as haptens and imitating transition states of catalytic reactions [1,2,3,4,5,6,7,8]. However, different anti-idiotypic autoantibodies can be induced against catalytic sites of enzymes, and they may also possess various catalytic activities [9,10]. Natural Abzs hydrolyzing DNA, RNA, polysaccharides [11,12,13], oligopeptides, and proteins [14,15,16] are described from the sera of patients with several autoimmune diseases. Sera of some healthy humans contain abzymes with low proteolytic [14,15] and polysaccharide-hydrolyzing activities [13]. Healthy volunteers and patients with some diseases with insignificant AI reactions usually lack abzymes with protease, RNase, and DNase activities [3,4,5,6,7,8]. Germline Abs from healthy humans can, however, express auto-Abs with promiscuous, amyloid-directed, and superantigen-directed activities and/or autoantigen-directed and microbe-directed specificities [17,18].

Multiple sclerosis (MS) is a chronic demyelinating disease of the central nervous system. Its etiology is unclear, and its widely accepted theory assigns the main role in the destruction of myelin due to AI reactions [19]. Several recent findings suggest an important role of B cells and auto-Abs against myelin autoantigens in MS pathogenesis [19,20,21]. Similarly to systemic lupus erythematosus (SLE), anti-DNA Abs were recently identified as a major component of the intrathecal IgGs in brains of MS patients and in cells of the cerebrospinal fluid [22]. It was shown that similarly to SLE, IgGs from the sera and the cerebrospinal fluid of MS patients were active in the hydrolysis of RNA, DNA, myelin basic protein (MBP), and polysaccharides [23,24,25,26,27]. In addition, the relative activities of IgGs from the cerebrospinal fluid of MS patients in the hydrolysis of MBP, DNA, and oligosaccharides, depending on their substrate, are about 40–60-fold higher than autoantibodies from the blood of the same patients [28,29,30]. In MS and SLE, anti-MBP abzymes with proteolytic activity can hydrolyze MBP of the myelin-proteolipid sheath of axons [28,29,30]. Since Abzs of MS patients [31], similarly to SLE patients [32], are cytotoxic and induce cell apoptosis; they can play a momentous role in MS and SLE pathogenesis. The question is, what other auto-Abs and abzymes can play an important role in the MS pathogenesis.

Many anti-DNA Abs of AI patients are directed against DNA-histones nucleosomal complexes appearing in the result of internucleosomal cleavage during cell apoptosis [33]. Apoptotic cells are the primary source of immunogens in SLE and other AI pathologies, and they are important in recognition, processing, and/or presentation of apoptotic auto-antigens by cells presenting antigens triggering AI processes [33].

Histones and their post-translational modifications can play an important role in chromatin remodeling and gene transcription. In addition to intranuclear functions, the histones can be pernicious after entering into the extracellular space [34]. The immunization of animals with exogenous histones leads to systemic inflammatory and toxic reactions. Treatment of mice with various anti-histone preparations protects them from lethal endotoxemia, sepsis, ischemia, reperfusion injury, injury, pancreatitis, peritonitis, stroke, coagulation, and thrombosis. Moreover, higher histones concentrations in blood affect several pathophysiological processes and the progression of pathologies, including AI diseases, inflammation, and cancer [34]. It was recently shown that IgGs from sera of HIV-infected patients effectively hydrolyzed from one to five different histones (H1, H2a, H2b, H3, and H4) [35]. In addition, anti-histones IgGs of experimental autoimmune encephalomyelitis (EAE) of mice also well hydrolyze all five histones [36]. Thus, extracellular histones and their complexes with DNA can, in principle, together anti-DNA, anti-MBP, and anti-polysaccharides induce the development of MS.

In this report, we use several different approaches to provide the first evidence that IgGs from MS patients can hydrolyze different human histones, which may be important for the development of this disease.

## 2. Material and Methods

### 2.1. Chemicals, Donors, and Patients

All chemicals were purchased from Pharmacia or Sigma. Near equimolar mixture of five homogeneous bovine histones (H1, H2a, H2b, H3, and H4) was perched from Sigma, (product number H9250). For the study, we selected 59 MS patients (38 women and 21 men; mean age = 36.2 ± 10 years) during the period from 2017 to May 2019, according to the classification of McDonald [37] and the the criteria for definite MS and admitted multiple sclerosis center (Novosibirsk Medical University, Novosibirsk, Russia). The disease severity of all MS patients was scored using Kurtzke’s Expanded Disability Status Scale (EDSS) [38] at the time of sample collection. We have analyzed eight patients with debut of multiple sclerosis (DMS; corresponding to the first coming of patients to the clinic for research after the early manifestations of signs of this pathology), 37 patients with remitting multiple sclerosis (RMS), 11 patients with secondary progressive multiple sclerosis (SPMS), and three patients with remittent progressive multiple sclerosis (RPMS). More detailed data are given in Appendix A. At entry, the patients had no fever or symptoms of acute infections. Moreover, none of the patients at the time of sample collection had not received any anti-MS therapies during the 6 months before the study.

The blood sampling protocols conformed by the local center human ethics committee guidelines (Ethics committee of Novosibirsk State Medical University, Novosibirsk, Russia). Medical University ethics committee specifically approved this study) in accordance with Helsinki ethics committee guidelines. All patients gave written consent to the presenting their blood for scientific purposes.

### 2.2. Antibody Purification

Electrophoretically and immunologically homogeneous polyclonal IgGs were obtained from sera of MS patients by coherent affinity chromatography of the sera proteins on protein G-Sepharose and following fast protein liquid gel filtration chromatography (FPLC) on a Superdex 200 HR 10/30 column as in [15,16,23,24,25,26,27,28,29,30,35]. The blood sera were additionally subjected to centrifugation (2000 rpm, 10 min) and then loaded on a protein G-Sepharose column (5 mL) equilibrated in the TBS buffer (0.15 M NaCl, 20 mM Tris-HCl, pH 7.5). The column then was washed using TBS to zero optical density. Non-specifically adsorbed proteins were eluted from sorbent by the same buffer containing 1% Triton X-100 and 0.5 M NaCl. The total IgGs fraction was specifically eluted using 50 mM glycine-HCl (pH 2.6). The fractions obtained were collected to tubes containing 50 μL of 0.5 M Tris-HCl (pH 8.5) and then each fraction was neutralized additionally using this buffer and dialyzed against Tris-HCl (10 mM, pH 7.5) containing 0.1 M NaCl. The protein of the central part of IgGs peak was concentrated and used in further purification.

IgGs were purified additionally (BioCAD workstation; Applied Biosystems, Foster City, CA, USA) by FPLC gel filtration on Superdex 200 HR-10/30 column (GE Healthcare, New York, NY, USA), equilibrated with TBS as described previously [25,26,27,28,29,30,35]. Prior to gel filtration, the IgGs samples were preincubated in TBS containing 2.5 M MgCl_2_ for 30 min at 20 °C. TBS containing 3 M MgCl_2_ (3 mL, “salt cushion”) was applied to the column before the samples. The IgGs were eluted using TBS. To protect IgGs from possible bacterial contamination, the preparations were filtered through a Millex filter (pore size 0.1 μm). After two weeks of storage (4 °C) for refolding the IgGs were used for the analysis of their protease activity as described below.

### 2.3. ELISA of Anti-Histones Autoantibodies

The relative concentration of anti-histones (sum of all five histones) auto-IgGs in total purified individual polyclonal IgG preparations of MS patients were measured using anti-histones ELISA test system from EUROIMMUN (Luebeck, Germany) according to the manufacturer’s instruction [35]. Bovine histones (equimolar mixture of H1, H2a, H2b, H3, and H4) were added to ELISA strips for overnight at 4 °C. The preparations of IgGs were used in the concentration 0.01 mg/mL. 50 μL citric-phosphate buffer containing 3,3′,5,5′-tetramethylbenzidine and H_2_O_2_ was added to the strips, mixtures incubated for 15 min at 22 °C. The reaction was stopped by the addition of H_2_SO_4_ (50 μL of 50%); the optical density (A_450_) was determined using Uniskan II plate reader (MTX Lab Systems, Bradenton, FL, USA). The concentrations of Abs against histones were expressed using the difference in the relative 450 nm absorbance (average of three measurements) between the samples containing and not containing Abs.

### 2.4. Ab proteolytic Activity Assay

The reaction mixture (10–60 μL) containing 20 mM Tris-HCl, pH 7.5, 1.0 mg/mL standard mixture of all five histones or recombinant homogeneous H1, and 0.01–0.2 mg/mL of polyclonal IgGs was incubated for 2–24 h at 37 °C as in [35]. After SDS-polyacrialamide elecnrophoresis (SDS-PAGE), the gels were analyzed by scanning and quantified using GelPro v3.1 software (Media Cybernetics, LP, USA). To quantitatively estimate the proteolytic activity, we have found a special concentration for each IgG preparation corresponding to the reaction of the pseudo first-order (linear part of the dependence of the hydrolysis rate upon [IgGs]; ([Abzs] << [S]) where histones are converted into their fragmented forms during incubation within the linear regions of the time dependencies. The relative activity (RA) was estimated from the decrease of the proteins in the band corresponding to each of five intact non-hydrolyzed histones.

The pH dependencies were analyzed using various buffer systems (50 mM): MES-NaOH (pH 5.3–6.6), Tris-HCl (pH 6.0–8.6) and glycine-NaOH (pH 9.0–10.0) as in [35]. To analyze the effect of ions of various metals, IgGs were used before and after three-time dialysis against 20 mM Tris HCl (pH 7.5) containing 0.1 M EDTA [35]. Then the solutions were passed through a Chelex 100 column. To determine the dependence of RAs on metal ions, IgG preparations (0.1 mg/mL) were preincubated with metal ions (CuCl_2_, NiCl_2_, MgCl_2_, MnCl_2_, FeCl_2_, FeCl_3_, CoCl_2_, ZnCl_2_, AlCl_3_, and CaCl_2_) at 4 mM concentration during 30 min at room temperature and then added to the standard reaction mixture (final concentration of metal ions 2 mM). The reaction products were analyzed by SDS-electrophoresis, as described above.

In some experiments IgGs (0.5–1 mg/mL) were preincubated for 30 min at 25 °C with one of specific inhibitors of various proteases: iodoacetamide (1.5 mM), phenylmethanesulfonylfluoride (PMSF, 1.0 mM), and EDTA (0.1 M), Pepstatin A (0.7 mM) as in [35]. Then, aliquots of the mixtures were added to the standard reaction mixture as in [35]. Since all initial rates of the reaction were measured within the linear regions of the time courses and IgGs concentrations, the measured RAs were normalized to standard conditions.

### 2.5. SDS-PAGE Assay of Ab Proteolytic Activity

Analysis of Abs by SDS-PAGE was carried out in 4–15% gradient gels under nonreducing conditions (0.1% SDS), and the proteins were visualized by silver staining [15,16,25,26,35]. Before SDS-PAGE, IgGs (10–15 μg) were preincubated at 30 °C for 30 min in Tris-HCl (50 mM; pH 7.5), containing 1% SDS, and 10%. To restore the protease activity, SDS was removed by incubating the gel for 2 h at 22 °C with 4 M urea, and the gel was washed eleven times with water. Then 3–4 mm cross-sections of the gel longitudinal slices were cut out, incubated with 20 mM Tris-HCl (50 μL, pH 7.5), containing 5 mM MgCl_2_ and 1 mM EDTA for four-five days at 4 °C to allow protein eluting from the gel and refolding. The eluates were used for the protease activity assay as described above. Parallel longitudinal slices of the gel were used to detect the position of IgGs in the gel by Coomassie staining.

### 2.6. Determination of the Kinetic Parameters

Reaction mixture (10–20 μL) for determination of K_m_ and V_max_ (k_cat_) values contained standard reaction components, 0–25 μM histone H1, and 0.05 mg/mL one of IgGs. The products of the hydrolysis analyzed using SDS-PAGE. The values of K_m_ and V_max_ (k_cat_) were determined from the kinetic data by least-squares non-linear fitting (Microcal Origin v5.0 software) and presented as linear dependencies using a Lineweaver-Burk plot [39]. Errors in the values were within 10–20%.

### 2.7. Statistical Analysis

The results concerning ELISA, relative activity of Abs, their dependence on pH, metal ions and other data are reported as the mean and the standard deviation of at least 2–3 independent experiments for each sample of sera and IgG preparation. For checking of normality of values distribution law, the criterion of Shapiro-Wilk’s W test was used. Several of the sample sets did not correspond to the normal Gaussian distribution. Therefore, the correlation coefficients (CCs) of different values sets were evaluated using the non-parametric Spearman test. The differences between different groups IgG samples were estimated using the Mann-Whitney test, *p* < 0.05 was considered statistically significant. The median (M) and interquartile ranges (IQR), as well as average values, were estimated for all groups of IgGs.

## 3. Results

### 3.1. IgG Purification and Characterization

It is known, that the generation of autoantibodies to self-antigens including DNA and different proteins usually occurs not only in patients with viral, bacterial and autoimmune pathologies but also in healthy humans [3,4,5,6,7,8,40,41,42].

Electrophoretically and immunologically homogeneous polyclonal IgGs were purified from sera of MS patients by sequential chromatography on Protein G-Sepharose under conditions removing non-specifically bound proteins. Then, IgGs were additionally purified using FPLC gel filtration in the condition destroying immune complexes as in [15,16,25,26,35]. For the characterization of IgGs, we used individual IgGs and a mixture of equal amounts of 25 MS IgGs (IgG_mix_) having detectable or high activity in the hydrolysis of several histones. The homogeneity of the 150 kDa IgG_mix_ was confirmed by SDS-PAGE with following silver staining, which showed a single band under nonreducing conditions (Figure 1A).

### 3.2. Titers of IgGs to Different Histones

The obtained homogeneous IgG preparations were used to evaluate in them the content of anti-histones Abs. For a total group of 59 individual MS patients, the level of anti-histones IgGs varied in a broad range from 0.033 to 0.86 A_450_/mL (average value is 0.14 ± 0.11 A_450_/mL). The median (M = 0.12 A_450_/mL) and interquartile ranges (IQR = 0.064 A_450_/mL) of these values for total group were estimated. Thirteen patient with debut of MS demonstrated A_450_ units from 0.033 to 0.17 (average value 0.07 ± 0.04; M = 0.056, IQR = 0.040 A_450_/mL). Thirty-seven patients with remitting multiple sclerosis (RMS) were characterized by a change in titers from 0.054 to 0.86 and a higher average value 0.16 ± 0.13, as well as M = 0.14, IQR = 0.056 A_450_ units. The third group of patients with SPMS demonstrated A_450_ units from 0.04 to 0.13 and lower average value (0.1 ± 0.04 A_450_/mL) in comparison with RMS, but higher than that for debut of MS. And the fourth-smallest group of three patients with remittently progressive multiple sclerosis (RPMS) was characterized by A_450_ values from 0.1 and to 0.2 (average value is 0.14 ± 0.05; M = 0.13, IQR = 0.09 A_450_/mL). The level of IgGs against histones in IgG preparations from serum of healthy donors was previously evaluated and it was varied from 0.041 tom 0.053 A_450_ units (average value 0.045 ± 0.004; M = 0.044, IQR = 0.003 A_450_/mL).

The significance of differences in the relative A_450_ (R) in the case of four groups of patients was evaluated by the Mann-Whitney test.

### 3.3. Catalytic Activity and Application of the Strict Criteria

The main part of the purified homogeneous IgG preparations effectively hydrolyzed all five histones (Figure 1B) and individual recombinant histone H1 (Figure 1C).

For attributing of histone-hydrolyzing activity directly to MS IgGs, we applied strict criteria worked out previously [3,4,5,6,7,8,43]. Individual IgGs and IgG_mix_ were electrophoretically homogenous (Figure 1A); the complete absorption of MS IgG_mix_ hydrolyzing five histones by anti-IgG-Sepharose led to a disappearance of the activity in the solution; FPLC gel filtration of IgG_mix_ preincubated with acidic buffer (pH 2.6) did not result in a disappearance of proteolytic activity, which tracked exactly with IgG_mix_ (Figure 2A); after SDS-PAGE of IgG_mix_ proteolytic activity of possible proteases was detected using solutions obtained after extraction of proteins from many gel slices (Figure 2B). The detection of protease activity in the solutions corresponding to gel fragments containing only IgG_mix_, as well as the absence of other bands of the activity or protein (Figure 2B), procures direct evidence that MS IgGs possess histone-hydrolyzing activity and is not contaminated by impurities of any canonical proteases, which have significantly lower molecular masses: proteases 24–25 kDa and metalloproteases 50–60 kDa). Therefore, we have estimated the RA of different MS IgGs without further additional purification.

### 3.4. Estimation of the Relative Proteolytic Activity

We have estimated the proteolytic activity of 59 individual IgGs from the sera of MS patients. The RAs were calculated from the decrease of the histones in the bands corresponding to five initial non-hydrolyzed proteins. IgGs from the sera of various MS patients hydrolyzed all histones (H1, H2a, H2b, H3, and H4) demonstrating a very different efficiency. Therefore, to quantify the RAs in the hydrolysis of different histones by each IgG preparation, the time of incubation (8–24 h) and Abs concentrations (0.01–0.2 mg/mL) were significantly varied, but in all cases conditions of the pseudo first-order reaction were used (for example, Figure 1B,C). This allowed us to normalize the RAs to standard conditions (% of histone/0.05 mg/mL of IgGs/20 h); the data summarized in Table 1. Unfortunately, it was not possible by electrophoresis to ideally separate histones with comparable molecular masses. Therefore, the errors in average values based on three independent experiments (especially for IgGs with low activity) were relatively high and can reach up to 10–25%.

The RAs of IgGs for the whole group of 59 patients in the hydrolysis of various histones significantly varied (from 0 to 90%) from patient to patient. In addition, 15 of 59 preparations (25.4%) did not hydrolyze any of the five histones (Table 1). Only two IgGs (RMS12 and SPMS8) hydrolyze only one of two histones (H3 or H2b). All other 44 IgGs (74.6%) hydrolyze from two to five histones (Table 1). The RAs for individual Abs of whole group do not correspond to normal Gaussian distribution. We have, therefore, estimated average values ± SD, and also the median (M) and interquartile ranges (IQR). In overall, average RAs of 59 IgGs in the hydrolysis of five histones are to a certain degree comparable (%): H1 (18.6 ± 20.9; M = 11, IQR = 31), H2a (19.5 ± 24.8; M = 10, IQR = 37), H2b (19.4 ± 24.8; M = 11, IQR = 35), H3 (22.1 ± 24.8; M = 13, IQR = 35), and H4 22.5 ± 27.5; M = 10, IQR = 46), and all five histones (20.4 ± 24.5; M = 11, IQR = 36.8) (Table 1).

The correlation coefficients (CCs) of RAs all 59 IgGs in the hydrolysis of five different histones were evaluated according non-parametric Spearman test: H1–H2a (0.738), H1–H2b (0.798), H1–H3 (0.716), H1–H4 (0.645), H2a–H2b (0.890), H2a–H3 (0.726), H2a–H4 (0.704), H2b–H3 (0.704), H2b–H4 (0.607), and H3–H4. (0.632). The highest level of CC was observed for H2a–H2b (0.89), while the lowest for H2b–H4 (0.607).

The significance of differences in the relative activities of IgGs from 59 patients (*R*) in the hydrolysis of five different histones (H1–H4) was evaluated using the Mann-Whitney test. There is no found statistically significant difference in the hydrolysis of five histones (H1–H4) by the antibodies. The *R* values were higher 0.05 and varied in the range 0.408–0.984 (average value is 0.683 ± 0.179).

Abs of only one patient with debut of MS (DMS5) demonstrated hydrolysis of only two histones (H2a and H4), all other 6 IgGs show hydrolysis of all five histones (Table 1). Average RAs of IgGs from patients with debut of MS in the hydrolysis of five histones were (%): H1 (30.9 ± 27.7; M = 18.3, IQR = 48), H2a (33.0 ± 24.7; M = 20.0, IQR = 46.5), H2b (30.8 ± 28.3; M = 18, IQR = 53.5), H3 (38.6 ± 21.8; M = 37, IQR = 30), and H4 (45.8.5 ± 15.8; M = 47, IQR = 23.5). Average RA of IgGs of debut MS in the hydrolysis of all five histones (35.9 ± 23.4%) was 1.5-fold higher than that for IgGs of all 59 patients (20.4 ± 24.5%).

The 12 out of 15 IgGs which had no activity in the hydrolysis of all five histones belonged to 37 patients with remitting multiple sclerosis (RMS), and one preparation (RMS12) hydrolyzed only the H3 histone (Table 1). Average RAs of IgGs from patients with RMS in the hydrolysis of five histones were (%): H1 (16.0 ± 20.5; M = 10, IQR = 25), H2a (16.8 ± 26.0; M = 0, IQR = 27), H2b (16.8 ± 25.6; M = 0.0, IQR = 25), H3 (20.2 ± 26.0; M = 10, IQR = 35), and H4 (15.6 ± 26.3; M = 0.0, IQR = 24). Average RA of all IgGs of this group in the hydrolysis of all five histones (17.1 ± 24.9%) was 1.2- and 2.0-fold lower than that for IgGs of all 59 and debut MS patients, respectively.

Two IgGs of 11 preparations of patients with secondary progressive multiple sclerosis (SPMS7 and SPMS10) did not possess activity in hydrolysis of all five histones, while one other (SPMS8) cleaved only H2b histone (Table 1). Average RA values for 11 IgGs were calculated (%): H1 (20.9 ± 17.5; M = 26, IQR = 33), H2a (19.8 ± 18.9; M = 25.0, IQR = 38.0), H2b (20.1 ± 17.4; M = 19.0, IQR = 35.0), H3 (15.8 ± 13.2; M = 13.0, IQR = 30.0), and H4 (33.8 ± 29.4; M = 32.0, IQR = 53.0). Average RA in the hydrolysis of five histones was 22.3 ± 19.3%, which is comparable with that for IgGs of all 59 patients (20.4 ± 24.5%).

We had only three patients with remittenting progressive multiple sclerosis (RPMS). RPMS2 has no catalytic activity, while RPMS1 is very weakly hydrolyzed only H4 (Table 1). The RAs of corresponding IgGs in the hydrolysis of five histones varied from 0.0 to 74.0% (Table 1). Average RA values were estimated (%): H1 (14.7 ± 25.4; M = 0.0, IQR = 44.0), H2a (16.7 ± 28.9; M = 0.0, IQR = 50.0), H2b (18.3 ± 31.8; M = 0.0, IQR = 55.0), H3 (24.7 ± 42.7; M = 0.0, IQR = 74.0), and H4 (5.0 ± 5.0; M = 5.0, IQR = 10.0). Average RA in the hydrolysis of five histones is 15.9 ± 26.7%.

The differences between various groups of IgG samples corresponding to four types of MS patients in the hydrolysis of five various histones were estimated using the Mann-Whitney test. Statistically significant differences (R < 0.05) were observed for the following groups of RAs; DMS-RMS: H2a—0.021, H3—0.035, H4—0.001; DMS-SPMS: H3—0.034; DMS-RPMS: H4—0.019, and RMS-SPMS: H4—0.029. No significant difference in the hydrolysis of five histones between all four groups of MS patients was found, *R* values varied from 0.074 to 1.0.

We determined the total titers (A_450_/mL) of antibodies against five histones in the total preparations of polyclonal IgGs. To assess the CCs between titers of anti-histones IgGs and relative catalytic activity of abzymes, the activities of different individual IgGs in the hydrolysis of five histones were summarized and CCs between these values and A_450_ units were calculated. For a total group of 59 preparations, the CC was very low and equal to +0.02. For individual groups, the values of CC were as follows: DMS (+0.4), RMS (+0.13), SPMS (−0.63), and RPMS (+0.94).

For IgGs of 15 patients having no activity in the hydrolysis of any of the five histones, the titers of Abs against histones were not zero and varied from 0.12 to 0.18 A_450_ units (average value 0.13 ± 0.02 A_450_/mL). Interestingly, this value is higher than that for IgGs of DMS patients (0.07 ± 0.04 A_450_/mL) demonstrating maximal activity in the hydrolysis of all five histones. This is consistent with our previous data that, depending on the stage of different AIDs, there may be a different level of antibodies without catalytic activity and abzymes hydrolyzing different substrates.

### 3.5. Type of Proteolytic Activity

Most often, abzyme preparations against various proteins have serine protease activity, which is strongly reduced after their incubation with specific serine protease inhibitor AEBSF [1,2,3,4,5,6,7,8]. Recently we have revealed an important metal-dependent protein-hydrolyzing activity of IgGs from the sera of patients with several AI diseases, which activity was well suppressed by EDTA [5,6,7,8]. Similar to Abs from sera of AI patients with protease abzymes, IgGs and IgMs from HIV-infected patients hydrolyzing integrase were inhibited by specific inhibitors of serine and metal-dependent proteases, but significant inhibition of thiol-like protease activity with iodoacetamide was observed for the first time [44,45].

We have analyzed a possible type of histone-hydrolyzing activity of four IgG preparations of MS patients. The inhibition of histones hydrolysis catalyzing with four IgGs was very different; the RA of the hydrolysis without inhibitor was taken for 100% (Table 2). The average level of inhibition of different IgGs by AEBSF was varied in the following order (%): H1 (26–100); H2a (0–36); H2b (26–97) H4 (19–98) (Table 2). Effect of EDTA on the activity of various IgGs in the hydrolysis of the histones was also different (%): H1 (0–43); H2a (0–78); H2b (0–66) H4 (0–70).

Iodoacetamide, a specific inhibitor of thiol proteases, more often does not significantly suppress the activity of proteolytic abzymes (≤3–15% of inhibition) [5,6,7,8,15,16,25,26]. It is only AIDS IgGs, and IgMs hydrolyzing integrase could be suppressed by iodoacetamide for 12–98%, which is quite different comparing to other known Abs with proteolytic activity [44,45]. The data on the inhibition of four histones hydrolysis by iodoacetamide are summarized in Table 2. The inhibition of the activity of four different IgGs in the hydrolysis of the histones was varied in the following order (%): H1 (0–54), H2a (0–22), H2b (0–7), and H4 (16–41) (Table 2).

A specific inhibitor of acidic proteases (pepstatin A) demonstrated usually insignificant inhibition (4–5%) of proteolytic activity in the hydrolysis of different specific proteins by individual IgGs from patients with different AI pathologies [5,6,7,8,15,16,25,26]. A somewhat unexpected situation was revealed in the case of histones hydrolysis by IgGs from patients with MS (Table 2). Some IgGs demonstrated a high level of protease activity inhibition by pepstatin A. Especially strong inhibition of the hydrolysis of H2a, H2b, and H4 was observed for DMS6 and DMS8 (40–61%). This means that, in contrast to abzymes against other different proteins, Abs against histones more often can have active sites of acid proteases.

In overall, the average relative inhibition of different types of proteolytic activities of MS IgGs is reduced in the following order (%): AEBSF (54.7) > Pepstatin A (31.2) > EDTA (29.8) > iodoacetamide (15.9).

### 3.6. The Effect of External Metal Ions on the Activity of IgGs

It was shown that mammalian IgGs do not completely lose intrinsically bound metal ions during the standard procedure of IgG purification [46,47]. Similar to abzymes against different proteins [5,6,7,8,15,16,25,26], MS IgGs contain specific fractions dependent and independent on metal ions (Table 2). For the study of possible effects of ion metals on the histones-hydrolyzing activity, we have used two preparations (DMS6 and DMS8). The dialysis of IgGs results in an incomplete decrease in the activity of IgGs in the hydrolysis of different histones by DMS6 and DMS8; the decrease in the activity was estimated (-fold): H1 (5.0 and 1.7, respectively), H2a (1.1 and 2.4), H2b (5.8 and 4.0), H4 (3.6 and 1.4) (Figure 3). The different external metal ions’ effects on the hydrolysis of individual histones were different. However, maximal activation of the activity in the hydrolysis of all four histones by DMS6 and DMS8 preparations was observed for Al^3+^ and Fe^3+^. DMS8 was also well-activated by Ca^2+^ (Figure 3E–H), while these ions were inhibitor of DMS6 (Figure 3A–D). Ni^2+^ activated hydrolysis of H2a, H2b, and H4 histones by DMS6 (Figure 3B–D), while Cu^2+^ stimulated cleavage the same three histones with DMS8 (Figure 3F). Other metal ions showed significantly less activating or inhibiting effects, which are individual for all metal ions and IgG preparations.

Therefore, it should be noticed that for abzymes hydrolyzing histones from the blood HIV-infected patients was observed a completely different dependence on metal ions [44,45]. The maximal activation of HIV-infected patients IgGs were observed mainly in the presence of Cu^2+^, Mn^2+^, Co^2+^, Ni^2+^, and Zn^2+^. Thus, the ratio of RAs of IgGs from various MS patients before and after removing metal ions, and of their RAs in the presence of different metal ions, similar to abzymes against other proteins may be individual for every preparation analyzed. These data are indicative of Me^2+^ dependence diversity of IgGs from the sera of MS patients in the hydrolysis of five histones.

### 3.7. pH Dependencies of Histones Hydrolysis

All human proteases usually have only one pronounced pH optimum [39]. It was shown earlier that polyclonal Abs against various proteins are extremely heterogeneous and more often can hydrolyze proteins in pH range from 5 to 10 demonstrating from 2 to 7 pH optima [4,5,6,7,8]. Three preparations of MS Abs (DMS6, DMS7, and DMS8) catalyzed the hydrolysis of five histones in the wide range of pH from 5.5 to 10 (Figure 4). The profiles of these dependencies were substantially different. DMS6 hydrolyzes five histones with nearly the same intensity at pH from 5.5 to 10 with small peaks at pH close to 6, 7, and 8 (Figure 4A). DMS7 demonstrates three brightly pronounced peaks of the activity in the hydrolysis of five histones at pHs close to pH 5.5, 7.5, and 9.0 (Figure 4B). DMS8 shows only two relatively high peaks of the activity at pHs 5.5 and 8.0, and it is inactive in the pH range from 6.0 to 7.5 (Figure 4C). These results clearly demonstrate that, similar to abzymes against other different proteins [3,4,5,6,7,8], polyclonal IgGs from every MS patient may be very heterogeneous. They can demonstrate dependently of MS patients quite distinct individual pH dependencies in the hydrolysis of five histones.

### 3.8. Affinity of IgGs for Histones

For estimation of possible affinity of individual H1 histone to MS abzymes, four different preparations were used: DMS6, DMS7, DMS8, and SPMS1. All dependencies were not familiar with Michaelis-Menten kinetic; they have at least two or three parts of the dependencies (for example, Figure 5). It was difficult to determine K_m_ and k_cat_ values corresponding to the second part of such dependencies (Figure 5). Therefore, we evaluated these values only from the first part (for example, Figure 5B,D, and insert to Figure 5E).

For three preparations (DMS6, DMS7, and DMS8), the K_m_ values for histone H1 were comparable (µM): 4.5.0 ± 0.5 (DMS6), 6.7 ± 0.8 (DMS7), and 4.0 ± 0.5 (DMS8). The K_m_ values corresponding to the second parts of the curves for H1 and three IgGs (DMS6, DMS7, and DMS8) may be approximately estimated 2–7-fold higher than that for the first parts. Interestingly, SPMS1 demonstrated more complex dependence of the rate on the concentration of H1 corresponding to a sum of at least three hyperbolic curves of abzymes saturation with H1 histone and K_m_ values approximately equal to 4–6, 14–17, and >22–24 µM. The difference in the k_cat_ values for three preparations was also insignificant: 0.28 ± 0.04 min^−1^ (DMS6), (0.30 ± 0.04) min^−1^(DMS7), 0.18 ± 0.03 min^−1^ (DMS8). The k_cat_ values corresponding to the second parts of the dependencies in the case of DMS6, DMS7, and DMS8 may be also higher approximately by a factor of 2–7.

## 4. Discussion

We have analyzed previously a possible activity of ten polyclonal IgGs of healthy donors in the hydrolysis of five histones in comparison with 32 IgG preparations from HIV-infected patients [35]. It was shown that IgGs of ~50% of contingently healthy donors hydrolyze H1, H2a, H2b, H3, and H4 histones with detectable rate. The average activity in the hydrolysis of the histones by IgGs from HIV-infected patients was ~16-fold higher than that for healthy donors [35].

In this article, we analyzed the activity of 59 IgG preparations from the sera of MS patients. Interestingly, as in the case of healthy donors (50% of donors), but in contrast to HIV-infected patients (0% of patients), 15 of out 59 IgGs (~25.4%) from sera of MS patients cannot hydrolyze any of the histones, The average activity of all histones hydrolysis by IgGs of MS patients in terms of the same conditions was ~3.9-fold higher than that for healthy donors, but 6.1-fold lower than that for HIV-infected patients. This means that, overall, the development of MS leads to an increase in the number of people whose lymphocytes produce abzymes against five different histones more effectively that in healthy humans.

Interestingly, the relative activity of IgGs in the hydrolysis of five histones depends on the type of MS. The maximum activity of IgGs is observed in the debut of this pathology, which is characterized by relatively high average RA in the hydrolysis of all five histones (35.9%; Table 1). This value is 1.5-fold higher than that for IgGs, corresponding to all 59 patients of four MS groups (20.4%). 12 of 37 preparations of RMS (32.4%) have no activity in the hydrolysis of all five histones, while several others demonstrate relatively low activity (Table 1). The average RA of IgGs of this group (17.1%) is 1.2- and 2.1-fold lower than that for IgGs of all 59 and debut MS patients, respectively.

Average RA for 11 IgGs with secondary progressive multiple sclerosis in the hydrolysis of five histones is 22.3%, which is comparable with that for IgGs of all 59 patients (20.4%), but 1.6 lower than that for debut of MS patients.

There were only three patients with remitting progressive multiple sclerosis (RPMS) demonstrating average RA in the hydrolysis of five histones equal to 15.9%.

Thus, the maximal activity of IgGs in the hydrolysis of five histones is observed at the debut of this pathology. This is consistent with the data on the change in the activity of the abzymes hydrolyzing DNA, myelin basic protein (MBP), MOG (fragment of myelin oligodendrocyte glycoprotein), and histones after immunization of C57BL/6 mice prone to experimental autoimmune encephalomyelitis (EAE) with various antigens [36,48,49]. It was shown that IgGs from mice blood demonstrate maximal activity at the beginning and acute phase of EAE, while the activity decreased significantly later at the remission phase of EAE.

The CCs of RAs all 59 IgGs in the hydrolysis of five different histones relatively high and varying from 0.607 to 0.89. The affinity of IgGs for H1 histone in terms of K_m_ values (K_m_ = 4.0–6.7 μM) is comparable with typical affinities (K_m_ = 0.038–16 μM) [3,4,5,6,7,8,15,16,25,26] of abzymes to their specific protein-substrates.

The dialysis of IgGs leads to a decrease in their activity in the hydrolysis of five different histones 1.1–5.8-fold (Figure 3). Maximal activation of dialyzed IgGs was observed for Al^3+^ and Fe^3+^ ions. DMS8 was also well activated by Ca^2+^ (Figure 3E–H), while these ions were inhibitor for DMS6 (Figure 3A–D). The maximal activation of abzymes from the blood HIV-infected patients was observed mainly in the presence Cu^2+^, Mn^2+^, Co^2+^, Ni^2+^, and Zn^2+^ [44,45]. However, these ions demonstrated very different effects on the activity of MS IgGs in the hydrolysis of five histones, and some of them were inhibitors of their proteolytic activities (Figure 3). These data are indicative that Me^2+^ dependences of IgGs from the sera of patients with MS and other diseases in the hydrolysis of five histones may be different.

Proteolytic IgGs of patients with various AI pathologies are not usually only serine-like, but also thiol-like, metal-dependent, and acidic proteases [3,4,5,6,7,8]. The ratio of the RAs of human abzymes with different types of proteolytic activities is significantly dependent on AI pathology and the specific protein-substrate [3,4,5,6,7,8]. It was shown that 22 of 72 analyzed recombinant monoclonal light chains (MLChs) of SLE mice efficiently hydrolyzed only MBP [50]. Four of them have serine-like protease; three demonstrate thiol protease-like activity, while eleven MLChs were metalloproteases. The activities of three chimeric MLChs were suppressed by both PMSF and EDTA, and one by PMSF, EDTA, and iodoacetamide. It was shown that three MLChs demonstrated two pH optima, two K_m_ values for MBP and in their active sites are combined two active centers corresponding to either two different metalloproteases or a metalloprotease and serine-like protease [51,52,53]. The protein sequences of these MLChs, having two active centers in one site, demonstrate homology with several mammalian metalloproteases and/or serine-like proteases [51,52,53]. These data on polyclonal IgGs [3,4,5,6,7,8] and MLChs [50,51,52,53] suggest an extreme diversity of anti-protein abzymes in SLE and other autoimmune pathologies.

Similar situation was observed for MS IgGs hydrolyzing histones. The activity of IgGs was inhibited by PMSF, EDTA, iodoacetamide, and pepstatin A (Table 2). If the abzyme active center has only one type of proteolytic activity, then the maximum inhibition of this activity by one of four specific inhibitors should not exceed 100%. However, similar to polyclonal Abs from the sera of HIV-infected patients hydrolyzing HIV integrase [44,45], several IgG preparations of MS patients demonstrate serine-, thiol-like, metalloprotease, and/or acidic proteolytic activity: the sum of the effects of inhibition by several specific inhibitors exceeds 100% (Table 2). This is especially brightly observed for the following preparations: DMS6 (H2b and H4—212%), DMS7 (H4—207%), DMS8 (H2b—207 and H4—241%) (Table 2). Thus, some abzyme molecules have active centers with combined structural elements of several types of proteolytic enzymes.

IgGs possessing acidic proteolytic activity usually make a small contribution to the total abzyme activity hydrolyzing different proteins (4–5%) [3,4,5,6,7,8,15,16,25,26,28,44,45]. However, MS IgGs demonstrated a high level of histone-hydrolyzing activity suppressed by pepstatin A (40–61%) (Table 2). Thus, in contrast to abzymes against other different proteins, abzymes against histones can have active sites of acid proteases.

In contrast to all human proteases, having only one pronounced pH optimum [39], polyclonal abzymes against various proteins usually show several pH optima in the range of pH from 5 to 10 [3,4,5,6,7,8]. Most of twenty-five SLE monoclonal light chains hydrolyzed MBP demonstrating one specific pH optimum in the pH range from 5 to 10 [50], while three preparations have two pH optima [51,52,53]. Abzymes of MS patients also demonstrated several pH optima in the hydrolysis of the histones over the wide range of pH values (5.5–9.5) (Figure 5). Our data point to the fact that anti-histones abzymes of different MS patients can contain subfractions of monoclonal abzymes with very different pH optima.

## 5. Conclusions

Taken together, we for the first time have shown here that the sera of MS patients can contain abzymes hydrolyzing from one to five histones; the average relative activity of IgGs from the MS patients is significantly higher than that for the healthy volunteers.

## Figures and Tables

**Figure 1 biomolecules-09-00741-f001:**
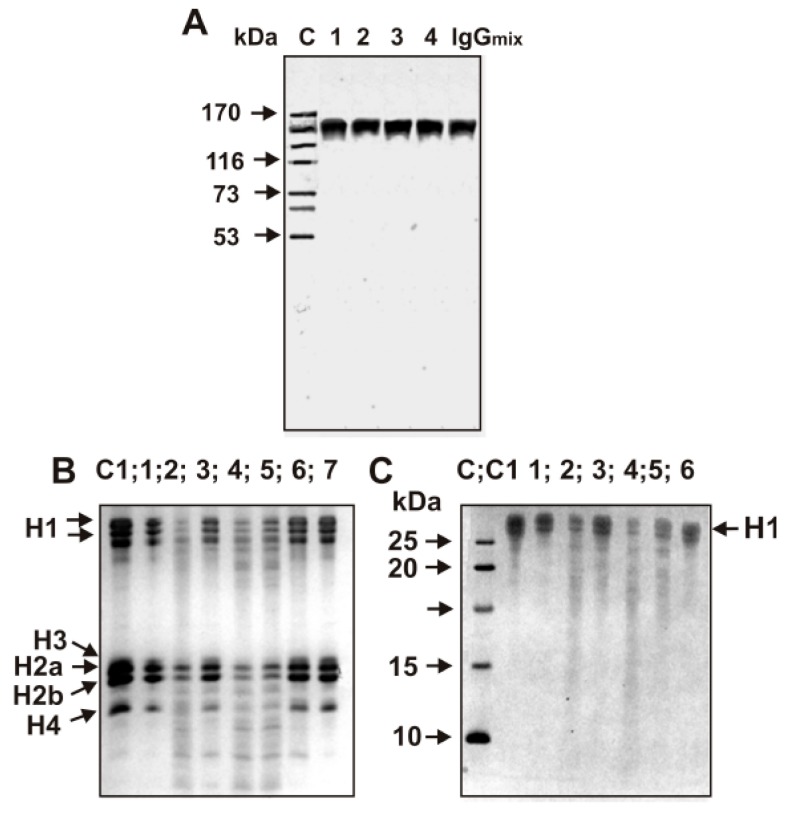
(**A**) SDS-PAGE analysis of homogeneity of four individual IgGs and IgG_mix_ (9 μg) from the sera of MS patients in a nonreducing 3–16% gradient gel (lanes 1–4 and IgG_mix_) followed by silver staining The arrows (lane C) indicate the positions of protein molecular mass markers; (**B**) SDS-PAGE analysis of the activity of several different IgGs in the hydrolysis of H1, H2a, H2b, H3, and H4 histones resulting in the formation of their fragments. The reaction mixtures was incubated for 20 h at 37 °C with 0.05 mg/mL IgGs (lanes: 1—RMS3, 2—debut of multiple sclerosis (DMS6), 3—secondary progressive multiple sclerosis (SPMS)1, 4—DMS7, 5—DMS8, 6—SPMS2, 7—RMS4); (**C**) Hydrolysis of recombinant H1 histone by several different IgGs leading to the formation of their fragments; lanes: 1—RMS3, 2—DMS6, 3—SPMS1, 4—DMS7, 5—DMS8, 6—RMS9 (Lanes C1 correspond to five histones (**B**) and H1 (**C**) incubated without Abs. (**C**) Lane C shows the position of protein markers with known molecular masses (**C**).

**Figure 2 biomolecules-09-00741-f002:**
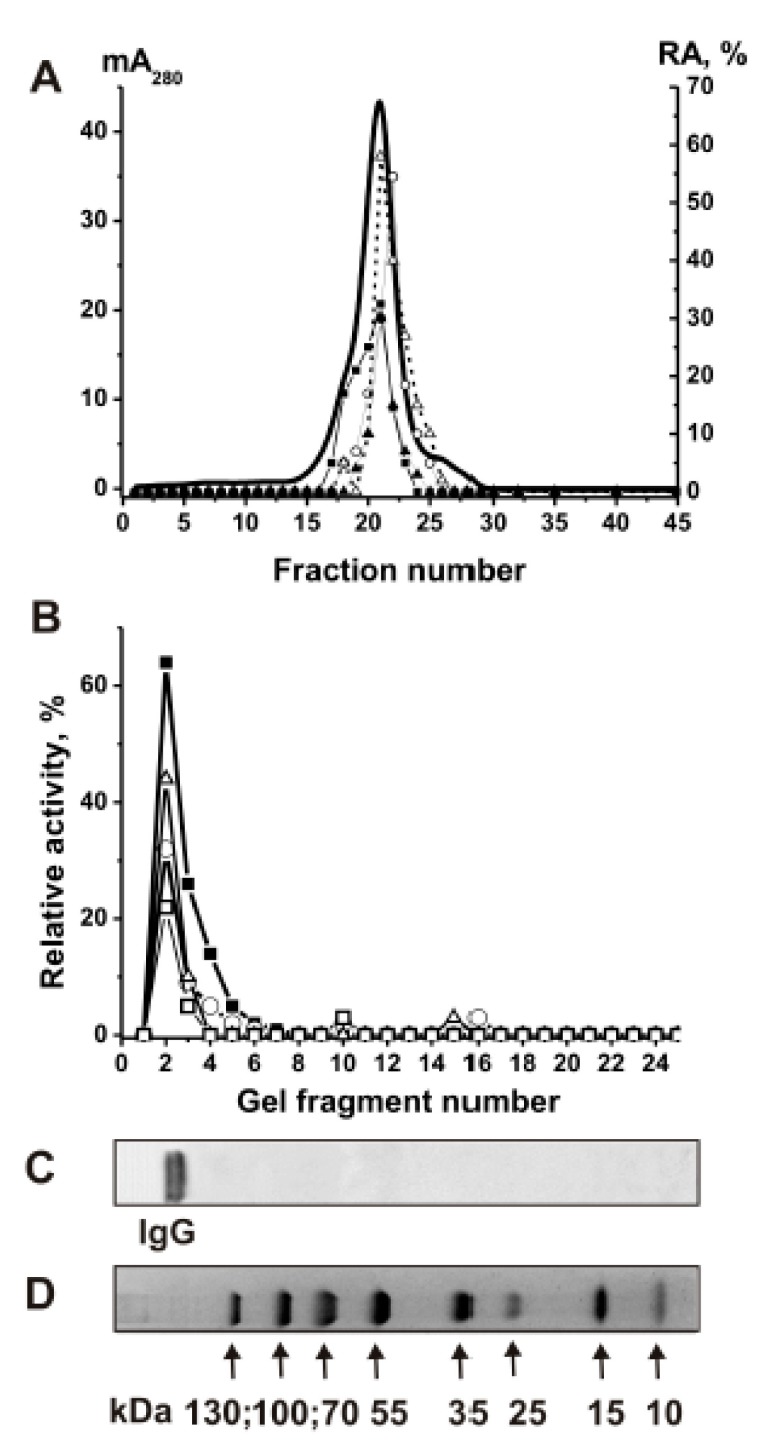
Verification of the strict criteria proving that IgG_mix_ hydrolyze histones. Fast protein liquid gel filtration chromatography (FPLC) gel filtration of MS IgG_mix_ on a Superdex 200 column in glycine buffer (pH 2.6) after Abs pre-incubation using the same buffer. (**A**) (—), Absorbance at 280 nm (A_280_); (■, ○, ∆, ▲) relative activities (RAs) of IgG_mix_ in the hydrolysis of H1, H2a, H2b, and H4, respectively. A complete hydrolysis of each histone for 16 h in the presence of 5 µl of the eluates was taken for 100%. SDS-PAGE analysis of IgG_mix_ in the hydrolysis of histones; (**B**) After non-reducing SDS-PAGE of MS IgG_mix_ in 3–15% gradient gel, it was incubated using special conditions for IgGs renaturation. The RA in hydrolysis of H1, H2a, H2b, H3, and H4 histones (■, ○, ∆, □, respectively) were estimated using the extracts of many 3–4 mm fragments of one longitudinal slice of the gel; (**C**) The RA of IgG_mix_ corresponding to a complete hydrolysis of every histone after 16 h of incubation with 5 μL of extracts was taken for 100%. The second control longitudinal slices of the same gels were stained with Coomassie R250; (**D**) Shows the position of protein with known molecular masses. The error in the RAs determination (two independent experiments) in hydrolysis of each histone did not exceed 10–20% (**A**,**B**). For details, see Materials and Methods.

**Figure 3 biomolecules-09-00741-f003:**
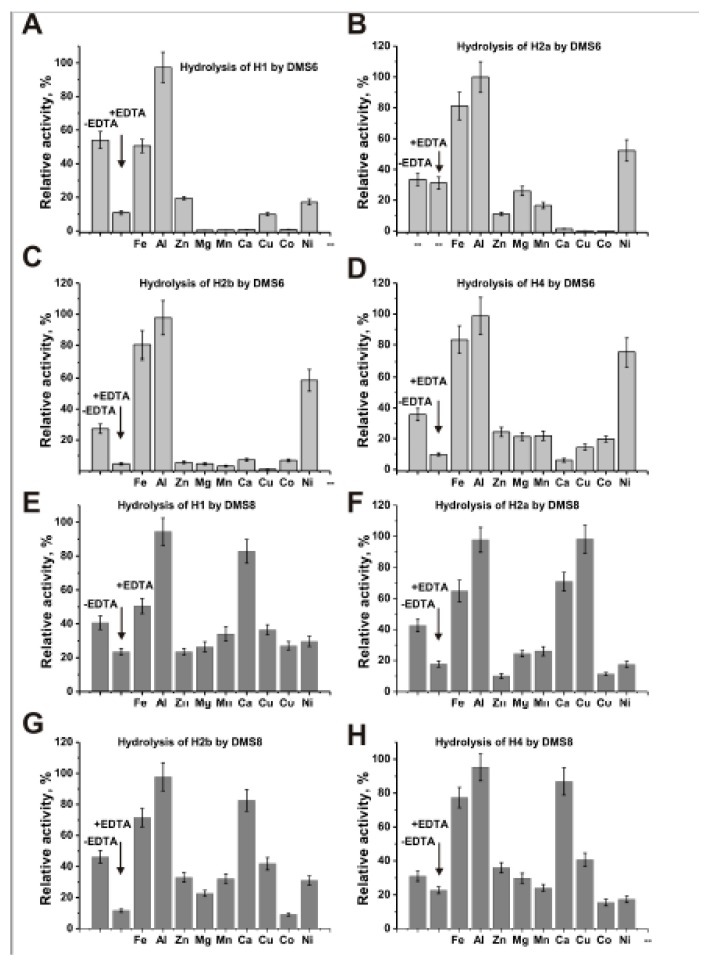
Typical examples of a decrease in histones-hydrolyzing activity of two individual IgGs (DMS6 and DMS8) after their dialysis against EDTA and then the activation of dialyzed IgGs by different metal ions at fixed 2 mM concentration (**A**–**H**). A complete hydrolysis of every histone in the presence of 0.66 μM IgGs for 12 h was taken for 100%. Diagrams corresponding to various metal ions are marked in panels **A**–**D** (DMS6) and **E**–**H** (DMS6); used metal ions are listed on these panels. The differences between different groups were estimated: the significant difference (*p* = 0.036) was found only between the hydrolysis of H4 by DMS8 and H2b by DMS6, in other cases, *p* was higher than 0.05. For details, see Materials and Methods.

**Figure 4 biomolecules-09-00741-f004:**
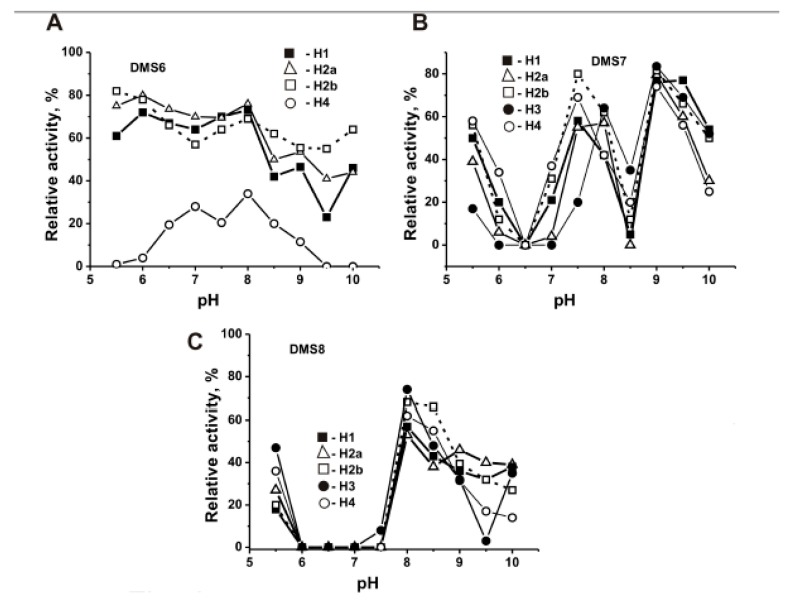
The pH dependence of the relative activity of three individual DMS6 (**A**), DMS7 (**B**), and DMS8 (**C**) preparations in the hydrolysis of the histones. The relative activity corresponding to a complete hydrolysis of every histone after 6 h of incubation in the presence of 0.33 μM IgGs was taken for 100%. The average in the initial rate determination error from two experiments did not exceed 10–25%. Significant difference (*p* < 0.05; 0.0001–0.041) was revealed for the following data in the hydrolysis of histones: DMS6 (H1–H4; H2a–H4; H2b–H4), DMS6–DMS7 (H2a; H4; H1–H2b; H1–H4; H2a–H2b; H2b–H4; H2a–H3; H2b–H4), and DMS6–DMS8 (H1, H2a, H2b, H1–H2a; H1–H2b;H1–H3; H1–H4; H2a–H2b; H2a–H3; H2a–H4; H2b–H3; H2b–H4). In the remaining cases, *p* > 0.05. See Materials and Methods for other details.

**Figure 5 biomolecules-09-00741-f005:**
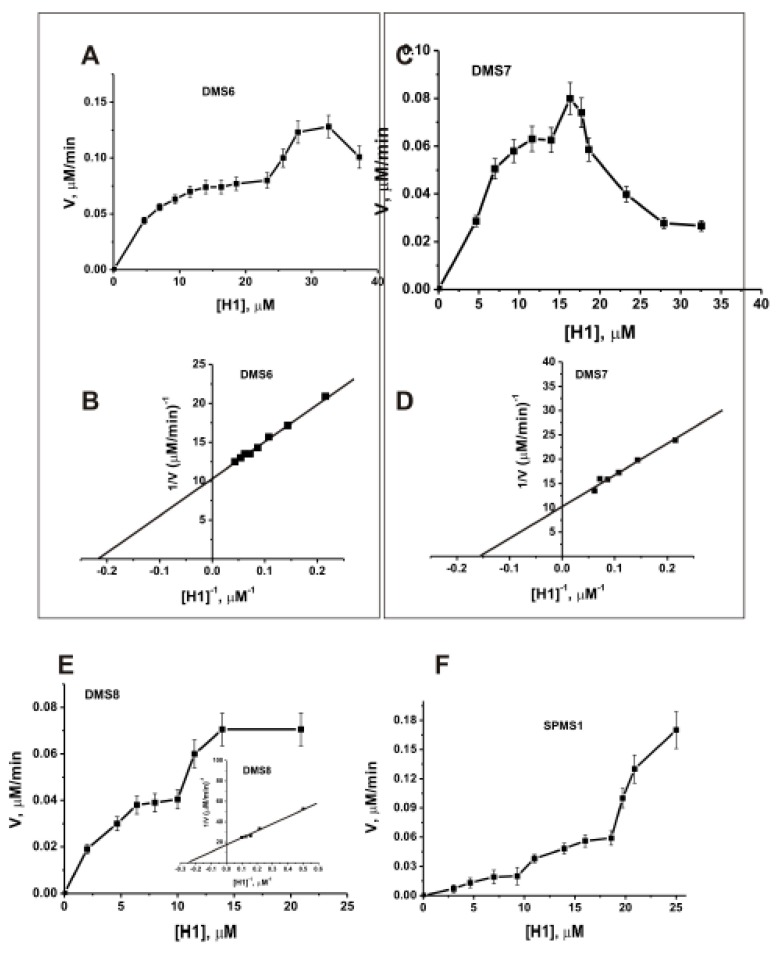
Dependencies of relative rates of H1 hydrolysis by different IgGs (0.33 μM): DMS6 (**A**,**B**), DMA7 (**C**,**D**), DMA8 (**E**), and SPMS1 (**F**) in different coordinates on the concentration of H1 histone. Determination of the K_m_ and V_max_ values for H1 histone in its hydrolysis by DMS6 (**B**) and DMA7 (**D**), and DMA8 (insert to **E**) was performed using the Lineweaver–Burk plot and the data corresponding to the first part of the hyperbolic curves of panels A and C, respectively; the insert to E corresponds to first part of the main dependence for DMA8. The error in the initial rate determination at each substrate concentration did not exceed 10–25%. Significance of differences between data of A and B panels, *p* = 0,0005. Reactions were performed as described in Materials and Methods.

**Table 1 biomolecules-09-00741-t001:** Relative activity of polyclonal IgGs from the sera of MS patients in the hydrolysis of five different histones.

Number	Denotation of Preparation	Relative Activity % *				
		H1	H2a	H2b	H3	H4
**Debut of Multiple Sclerosis (DMS) ****						
1	DMS1	22.0 ± 2.0	11.0 ± 1.5	21.0 ± 2.0	26.0 ± 3.5	48.0 ± 7.0
2	DMS2	11.0 ± 1.5	21.0 ± 2.0	5.0 ± 0.7	29.0 ± 3.3	36.0 ± 4.5
3	DMS3	14.5 ± 2.0	11.0 ± 1.4	15.0 ± 1.9	44.0 ± 4.9	46.0 ± 5.3
4	DMS4	5.0 ± 0.7	15.0 ± 1.8	13.0 ± 1.5	30.0 ± 4.0	35.0 ± 4.6
5	DMS5	0.0	19.0 ± 2.1	0.0	0.0	18.0 ± 2.5
6	DMS6	66.0 ± 6.0	57.0 ± 6.1	59.0 ± 6.2	51.0 ± 6.0	55.0 ± 7.0
7	DMS7	57.0 ± 7.0	68.0 ± 7.0	67.0 ± 7.1	64.0 ± 6.9	63.0 ± 7.3
8	DMS8	55.0 ± 6.0	62.0 ± 7.1	66.0 ± 7.0	65.0 ± 7.2	65.0 ± 6.9
Average Value		30.9 ± 27.7	33.0 ± 24.7	30.8 ± 28.3	38.6 ± 21.8	45.8 ± 15.8
Average Value for H1–H4		35.9 ± 23.4%				
Median (IQR)		18.25 (48.0)	20.0 (46.5)	18.0 (53.5)	37.0 (30.0)	47.0 (23.5)
**Remitting Multiple Sclerosis (RMS)**						
9	RMS1	0.0	0.0	0.0	39.0 ± 4.0	18.0 ± 2.5
10	RMS2	0.0	8.0 ± 1.0	0.0	16.0 ± 2.0	16.0 ± 2.2
11	RMS3	34.0 ± 4.0	37.0 ± 4.1	38.0 ± 4.5	26.0 ± 3.5	47.0 ± 5.2
12	RMS4	31.0 ± 4.2	40.0 ± 5.0	40.0 ± 5.4	36.0 ± 4.2	58.0 ± 6.9
13	RMS5	25.0 ± 3.0	20.0 ± 3.5	28.0 ± 2.0	29.0 ± 3.4	44.0 ± 5.1
14	RMS6	25.0 ± 3.2	28.0 ± 3.7	25.0 ± 3.4	44.0 ± 5.1	32.0 ± 4.0
15	RMS7	20.0 ± 2.9	27.0 ± 3.3	12.0 ± 1.9	17.0 ± 2.2	24.0 ± 3.1
16	RMS8	20.0 ± 2.0	20.0 ± 2.9	18.0 ± 2.8	33.0 ± 3.9	33.0 ± 4.0
17	RMS9	20.0 ± 3.4	45.0 ± 5.0	20.0 ± 3.1	43.0 ± 5.1	27.0 ± 3.2
18	RMS10	10.0 ± 2.0	0.0	40.0 ± 4.6	0.0	0.0
19	RMS11	40.0 ± 4.9	70.0 ± 8.0	80.0 ± 9.1	85.0 ± 9.3	0.0
20	RMS12	0.0	0.0	0.0	35.0 ± 4.3	0.0
21	RMS13	0.0	0.0	0.0	0.0	0.0
22	RMS14	0.0	0.0	0.0	0.0	0.0
23	RMS15	12.0 ± 1.7	0.0	0.0	10.0 ± 2.0	0.0
24	RMS16	9.0 ± 1.8	40.0 ± 4.8	40.0 ± 5.0	50.0 ± 6.2	10.0 ± 2.1
25	RMS17	5.0 ± 1.5	0.0	0.0	10.0 ± 1.7	10.0 ± 1.4
26	RMS18	10.0 ± 1.6	10.0 ± 1.5	0.0	10.0 ± 2.0	0.0
27	RMS19	0.0	0.0	0.0	0.0	0.0
28	RMS20	11.0 ± 2.1	0.0	0.0	5.0 ± 1.0	0.0
29	RMS21	0.0	0.0	0.0	0.0	0.0
30	RMS22	0.0	0.0	0.0	0.0	0.0
31	RMS23	0.0	0.0	0.0	0.0	0.0
32	RMS24	0.0	0.0	0.0	10.0 ± 2.0	0.0
33	RMS25	0.0	0.0	0.0	0.0	0.0
34	RMS26	0.0	0.0	0.0	0.0	0.0
35	RMS27	0.0	0.0	0.0	0.0	0.0
36	RMS28	0.0	0.0	0.0	0.0	0.0
37	RMS29	0.0	0.0	0.0	0.0	0.0
38	RMS30	26.0 ± 2.9	80.0 ± 9.1	90.0 ± 10.5	0.0	0.0
39	RMS31	0.0	0.0	0.0	0.0	0.0
40	RMS32	90.0 ± 10.0	90.0 ± 12.0	87.0 ± 9.9	90.0 ± 11.0	78.0 ± 9.8
41	RMS33	48.0 ± 6.0	76.0 ± 8.2	50.0 ± 6.3	90.0 ± 12.1	90.0 ± 11.0
42	RMS34	24.0 ± 3.1	20.0 ± 3.0	14.0 ± 2.4	0.0	90.0 ± 10.0
43	RMS35	55.0 ± 7.1	0.0	15.0 ± 2.1	40.0 ± 4.9	0.0
44	RMS36	47.0 ± 5.9	0.0	0.0	10.0 ± 1.7	0.0
45	RMS37	17.0 ± 2.2	18.0 ± 2.5	19.0 ± 2.3	20.0 ± 3.0	14.0 ± 2.0
Average Value		16.0 ± 20.5	16.8 ± 26.0	16.8 ± 25.6	20.2 ± 26.0	15.6 ± 26.3
Average Value for H1–H4		17.1 ± 24.9				
Median (IQR)		10.0 (25.0)	0.0 (27.0)	0.0 (25.0)	10.0 (35.0)	0.0 (24.0)
**Secondary Progressive Multiple Sclerosis (SPMS)**						
46	SPMS1	40.0 ± 5.4	48.0 ± 5.9	47.0 ± 5.6	35.0 ± 4.2	53.0 ± 6.4
47	SPMS2	32.0 ± 4.1	38.0 ± 4.4	35.0 ± 4.7	32.0 ± 4.9	51.0 ± 6.9
48	SPMS3	30.0 ± 3.9	33.0 ± 4.0	32.0 ± 4.3	30.0 ± 4.2	58.0 ± 7.0
49	SPMS4	33.0 ± 4.4	39.0 ± 4.5	43.0 ± 5.8	13.0 ± 1.9	50.0 ± 6.8
50	SPMS5	26.0 ± 3.2	25.0 ± 3.9	24.0 ± 3.5	18.0 ± 2.1	28.0 ± 3.4
51	SPMS6	24.0 ± 3.7	30.0 ± 4.1	19.0 ± 2.6	25.0 ± 2.9	32.0 ± 4.1
52	SPMS7	0.0	0.0	0.0	0.0	0.0
53	SPMS8	0.0	0.0	10.0 ± 2.0	0.0	0.0
54	SPMS9	0.0	5.0 ± 1.5	11.0 ± 2.1	11.0 ± 2.2	10.0 ± 1.9
55	SPMS10	0.0	0.0	0.0	0.0	0.0
56	SPMS11	45.0 ± 5.3	0.0	0.0	10.0 ± 2.3	90.0 ± 12.0
Average value		20.9 ± 17.5	19.8 ± 18.9	20.1 ± 17.4	15.8 ± 13.2	33.8 ± 29.4
Average value for H1–H4		22.3 ± 19.3				
Median (IQR)		26.0 (33.0)	25.0 (38.0)	19.0 (35.0)	13.0 (30.0)	32.0 (53.0)
**Remittently Progressive Multiple Sclerosis (RPMS)**						
57	RPMS1	0.0	0.0	0.0	0.0	5.0 ± 1.0
58	RPMS2	0.0	0.0	0.0	0.0	0.0
59	RPMS3	44.0 ± 5.9	50.0 ± 7.2	55.0 ± 6.8	74.0 ± 8.9	10.0 ± 2.1
Average Value		14.7 ± 25.4	16.7 ± 28.9	18.3 ± 31.8	24.7 ± 42.7	5.0 ± 5.0
Average Value for H1–H4		15.9 ± 26.7				
Median (IQR)		0.0 (44.0)	0.0 (50.0)	0.0 (55.0)	0.0 (74.0)	5.0 (10.0)
Average Value for 59 patients		18.6 ± 20.9	19.5 ± 24.8	19.4 ± 24.8	22.1 ± 24.8	22.5 ± 27.5
Average Value for H1–H4		20.4 ± 24.5				
Median (IQR) for 59 Patients		11.0 (31.0)	10.0 (37.0)	11.0 (35.0)	13.0 (35.0)	10.0 (46.0)

* The data normalize to standard conditions (% of 1 mg/mL of five histones hydrolyzed/0.05 mg/mL of IgGs/20 h). Average values are reported as mean ± S.E., for each value, a mean of three measurements is reported; the error of the determination of values did not exceed 10–25%. ** Debut of multiple sclerosis (DMS) corresponds to the first coming of patients to the clinic for research after the early manifestations of signs of this pathology.

**Table 2 biomolecules-09-00741-t002:** Relative suppression of polyclonal IgGs activity from the sera MS patients in the hydrolysis of five different histones by specific inhibitors of proteases of different types *.

Denotation of IgG Preparation	Histones	Percent of Inhibition of Different Histones Hydrolysis **				
		EDTA 0.1 M	AEBSF	Iodoacetamide	Pepstatin A	Sum of the Effects
DMS6	H1	27.0 ± 2.0	96.0 ± 10.0	0.0	0.0	123.0
	H2a	20.0 ± 1.9	23.0 ± 2.6	10.0 ± 1.4	43.0 ± 4.7	96.0
	H2b	48.0 ± 5.2	97.0 ± 11.0	8.0 ± 1.2	54 ± 6.0	212.0
	H4	70.0 ± 8.1	58.0 ± 6.2	23.0 ± 2.9	61 ± 7.0	212.0
SPMS1	H1	0.0	52.0 ± 6.1	20.0 ± 2.0	15.0 ± 2.0	87.0
	H2a	0.0	36.0 ± 4.4	22.0 ± 2.8	54.0.0 ± 6.1	112.0
	H2b	0.0	62.0 ± 6.9	5.0 ± 1.0	59.0 ± 6.4	60.0
	H4	0.0	19.0 ± 2.5	41.0 ± 5.1	0.0	60.0
DMS7	H1	43.0 ± 5.0	26.0 ± 3.2	12.0 ± 1.9	0.0	81.0
	H2a	54.0 ± 6.0	0.0	0.0	0.0	54.0
	H2b	65.0 ± 7.2	43.0 ± 5.9	7.0 ± 1.2	9.0 ± 1.3	124.0
	H4	65.0 ± 7.3	54.0 ± 6.6	36.0 ± 4.2	52.0 ± 6.6	207.0
DMS8	H1	32.0 ± 4.4	100.0 ± 5.2	54.0 ± 6.7	0.0	186.0
	H2a	78.0 ± 9.2	22.0 ± 3.3	0.0	40.0 ± 5.2	140.0
	H2b	66.0 ± 7.7	89.0 ± 10.2	0.0	54.0 ± 6.7	209.0
	H4	69.0 ± 7.2	98.0 ± 11.0	16.0 ± 1.9	58.0 ± 7.4	241.0
Average Values		29.8 ± 28.8	54.7 ± 33.0	15.9 ± 16	31.2 ± 26.1	137.8 ± 64.1

* To estimate the inhibitory effects of all specific inhibitors The RAs of IgGs in the absence of inhibitors was taken for 100%. ** Average values are reported as mean ± S.E, for each value, a mean of three measurements is reported; the error of the determination of values did not exceed 10–25%.

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
