# Peer review of "Antibodies from the Sera of Multiple Sclerosis Patients Efficiently Hydrolyze Five Histones"

_biomolecules, 2019, doi:10.3390/biom9110741_

Round 1
Reviewer 1 Report
This is well written manuscript on effect of antibodies of multiple sclerosis (MS) patients on hydrolysis of histones. There are several minor points to address:
Figure 3,4,5, please add statistical analysis with p-values-when several groups compared. Western blot better add names of lanes above the blot rather than numbers-it will be easier for reader.
Author Response
Reviewer 1:
This is well written manuscript on effect of antibodies of multiple sclerosis (MS) patients on hydrolysis of histones. There are several minor points to address:
Figure 3,4,5, please add statistical analysis with p-values-when several groups compared. Western blot better add names of lanes above the blot rather than numbers-it will be easier for reader.
Answer:
Sorry, but there are eight panels in Fig. 3 that correspond to 28 values of P. It is completely unclear how they can be placed on eight panels. With this in mind, we gave values characterizing the significance of differences in the signature to the Figure.
Similar situation concerning Figure 4, where more than 50 values of P correspond to all the curves. Considering this, we gave values characterizing the significance of differences in the caption to the figure.
In Fig. 5 for generality, the values of P were also given in the caption to the Figure
Sorry, but the names are so long that they do not fit within the picture. Still, the numbers are left, there is no other way.

Reviewer 2 Report
Dear Editor
The manuscript by Baranova et al. reports that abzymes from the sera of multiple sclerosis (MS) patients hydrolyze human histones compared to those from healthy volunteers. This work is a follow up for an earlier study where the authors studied sera from AIDS patients.
The design of the study and the technical quality of the work are convincing and results can be of general interest. Authors used a good number of patients and the manuscript is well-written and easy to follow. The manuscript includes a detailed methodology that would enhance the reproducibility of the presented work. Data have been presented in good way and authors used correct statistical approaches in analyzing the results. Authors have successfully managed to discuss the findings of their study through an unbiased comparison literature.
However, there is a number of minor points that would need to be addressed in order to improve the quality of this paper before it can be accepted for publication:
Title: It shouldn’t have the word “all” since the data indicates that sera from MS patients can contain abzymes hydrolyzing 1-5 histones. Abstract: typo: after enter, it should be either following an entry or after entering. Introduction line 35: please rephrase as the presence of “to” twice makes the sentence hard to follow. Introduction line 36: typo: it should be written as “during the last three decades”. Introduction line 41: typo: it might be better written as anti-idiotypic. Introduction lines 76-77: rephrase to “higher histones concentrations in blood”. M&M line 90: typo: it should be “purchased”. M&M line 92: Define abbreviations whenever they appear first in the manuscript and use them throughout. Multiple Sclerosis to MS. M&M line 101: what do you mean by “anti-disease therapies”? Therapies or therapeutics should be enough unless you mean something more specific which isn’t clear. Statistical analysis line 183: Authors need to clearly specify which experiments have been done in two independent experiments and not with at least three replicates. Results line 368: typo: ) instead of ]. Discussion line 488: typo: this shouldn’t be in bold and font is off as well.
Best
Author Response
Reviewer 2:
The manuscript by Baranova et al. reports that abzymes from the sera of multiple sclerosis (MS) patients hydrolyze human histones compared to those from healthy volunteers. This work is a follow up for an earlier study where the authors studied sera from AIDS patients.
The design of the study and the technical quality of the work are convincing and results can be of general interest. Authors used a good number of patients and the manuscript is well-written and easy to follow. The manuscript includes a detailed methodology that would enhance the reproducibility of the presented work. Data have been presented in good way and authors used correct statistical approaches in analyzing the results. Authors have successfully managed to discuss the findings of their study through an unbiased comparison literature.
However, there is a number of minor points that would need to be addressed in order to improve the quality of this paper before it can be accepted for publication:
Title: It shouldn’t have the word “all” since the data indicates that sera from MS patients can contain abzymes hydrolyzing 1-5 histones.It was corrected:”Antibodies from the sera of multiple sclerosis patients efficiently hydrolyze five human histones”
Abstract: typo: after enter, it should be either following an entry or after entering.
Answer:
Introduction line 35: please rephrase as the presence of “to” twice makes the sentence hard to follow. Introduction line 36: typo: it should be written as “during the last three decades”. Introduction line 41: typo: it might be better written as anti-idiotypic. Introduction lines 76-77: rephrase to “higher histones concentrations in blood”. M&M line 90: typo: it should be “purchased”. M&M line 92: Define abbreviations whenever they appear first in the manuscript and use them throughout. Multiple Sclerosis to MS. M&M line 101: what do you mean by “anti-disease therapies”? Therapies or therapeutics should be enough unless you mean something more specific which isn’t clear. Statistical analysis line 183: Authors need to clearly specify which experiments have been done in two independent experiments and not with at least three replicates. Results line 368: typo: ) instead of ]. Discussion line 488: typo: this shouldn’t be in bold and font is off as well.
Answer:
All errors according to remarks 2-10 have been corrected.
Sincerely
Prof. Georgy Nevinsky

Reviewer 3 Report
The manuscript by Baranova et al assesses the ability of IgG purified from the sera of multiple sclerosis (MS) patients to hydrolyze bovine histones. Sera from 59 MS patients were studied, with a range of disease presentations. The authors revealed that the majority of purified IgG studied had the ability to hydrolyze at least one histone, and the pH dependency, and the ability of several proteases to inhibit hydrolysis, were examined.
The manuscript is well designed and the data clearly presented. I have a number of minor comments:
The title states that sera hydrolyze human histones, but I can only find reference to bovine histones in the Materials and Methods. If these were the only histones evaluated, please modify the title and manuscript to reflect this. Why were bovine histones chosen, and have the authors verified that these cross react with human IgG and in the ELISA assay that was carried out? Have the authors evaluated human histones in these assays, and do they expect that the results would differ from those obtained using bovine histones? In the discussion, the authors state that they previously analysed sera from healthy donors and HIV-infected patients for their ability to hydrolyse histones. Did they use the same experimental design for these studies? A limitation of the present study is the lack of uninfected control sera or sera from another inflammatory disease, and these data would have strengthened the manuscript.Author Response
Reviewer 3:
The manuscript by Baranova et al assesses the ability of IgG purified from the sera of multiple sclerosis (MS) patients to hydrolyze bovine histones. Sera from 59 MS patients were studied, with a range of disease presentations. The authors revealed that the majority of purified IgG studied had the ability to hydrolyze at least one histone, and the pH dependency, and the ability of several proteases to inhibit hydrolysis, were examined.
The manuscript is well designed and the data clearly presented. I have a number of minor comments:
The title states that sera hydrolyze human histones, but I can only find reference to bovine histones in the Materials and Methods. If these were the only histones evaluated, please modify the title and manuscript to reflect this. Why were bovine histones chosen, and have the authors verified that these cross react with human IgG and in the ELISA assay that was carried out? Have the authors evaluated human histones in these assays, and do they expect that the results would differ from those obtained using bovine histones?
Answer: It is known that histones of all mammals, unlike some other proteins, have a very high level of homology. Previously, we used human and bovine histones. It has been shown that they are absolutely equally hydrolyzed by abzymes from human blood. However, human histones are not sold by any of the firms. Previously, we obtained them ourselves, but this is a very laborious work. Only bovine histones are readily available. Since we did not find a difference in the hydrolysis of human and bovine histones, we began to use commercial preparations of bull histones.
In the discussion, the authors state that they previously analysed sera from healthy donors and HIV-infected patients for their ability to hydrolyse histones. Did they use the same experimental design for these studies?
Yes we have used the same experimental design for analysis of histones hydrolysis vy anzymes of Hiv infected and MS pdtients.
A limitation of the present study is the lack of uninfected control sera or sera from another inflammatory disease, and these data would have strengthened the manuscript.
Answer: Each study of abzymes from the blood of patients with various diseases in terms of work is very large and about the same as a study of catalytic antibodies from the blood of patients with multiple sclerosis. This does not fit in one article. So far, we have used antibodies from healthy donors as a control. Currently, work is underway with abzymes from the blood of patients with systemic lupus erythematosus. These data, as well as the results of the analysis of histone-hydrolyzing antibodies from the blood of patients with other diseases, will be published later. At this time, we have not collected a sufficiently large number of samples for a good statistical analysis.
Sincerely
Prof. Georgy Nevinsky
